# Cellular conditions of weakly chelated magnesium ions strongly promote RNA stability and catalysis

Ryota Yamagami[1,2], Jamie L. Bingaman[1,2,4], Erica A. Frankel[1,2,5] & Philip C. Bevilacqua[1,2,3]

Most RNA folding studies have been performed under non-physiological conditions of high concentrations ($\geq$10 mM) of $Mg^{2+}_{free}$, while actual cellular concentrations of $Mg^{2+}_{free}$ are only ~1 mM in a background of greater than 50 mM $Mg^{2+}_{total}$. To uncover cellular behavior of RNA, we devised cytoplasm mimic systems that include biological concentrations of amino acids, which weakly chelate $Mg^{2+}$. Amino acid-chelated $Mg^{2+}$ (aaCM) of ~15 mM dramatically increases RNA folding and prevents RNA degradation. Furthermore, aaCM enhance self-cleavage of several different ribozymes, up to 100,000-fold at $Mg^{2+}_{free}$ of just 0.5 mM, indirectly through RNA compaction. Other metabolites that weakly chelate magnesium offer similar beneficial effects, which implies chelated magnesium may enhance RNA function in the cell in the same way. Overall, these results indicate that the states of $Mg^{2+}$ should not be limited to free and bound only, as weakly bound $Mg^{2+}$ strongly promotes RNA function under cellular conditions.

[1] Department of Chemistry, Pennsylvania State University, University Park, PA 16802, USA. [2] Center for RNA Molecular Biology, Pennsylvania State University, University Park, PA 16802, USA. [3] Department of Biochemistry and Molecular Biology, Pennsylvania State University, University Park, PA 16802, USA. [4]Present address: Immunome, Inc., 665 Stockton Dr., Suite 300, Exton, PA 19341, USA. [5]Present address: The Dow Chemical Company, 400 Arcola Road, Collegeville, PA 19426, USA. Correspondence and requests for materials should be addressed to P.C.B. (email: pcb5@psu.edu)

RNA molecules fold into unique three dimensional structures, which allow them to function as catalysts that perform biochemical reactions and regulators that control protein expression[1]. Folding mechanisms, therefore, play significant roles in RNA function and related biological phenomena. In addition to primarily "classical standard in vitro conditions", which are high monovalent and divalent ion concentrations (e.g. ~1 M $Na^+$ and ~10 mM $Mg^{2+}$), the folding mechanisms of various RNA species have been studied in cellular mimicked conditions induced by cosolutes, and it is widely accepted that such crowding conditions increase ribozyme folding and activity[2–9].

In a recent metalome studies in *Escherichia coli*, total concentrations of potassium and magnesium ions were found to be relatively high, at 140–160 mM and 54 mM, respectively (Supplementary Fig. 1)[9–11]. However, free concentrations of magnesium, meaning fully hydrated by water at the inner-coordination sites on $Mg^{2+}$, are only ~0.5–1 mM in eukaryotic cells and ~1.5–3 mM in bacterial cells since most magnesium in the cytoplasm interacts with metabolites, proteins, and nucleic acids[6,9]. Recent metabolome analyses report greater than one hundred millimolar of amino acids, which account for ~50% of all metabolites in *E. coli* (Supplementary Fig. 1 and Supplementary Table 1)[12,13]. Thus, RNA is exposed to large amounts of amino acid-chelated magnesium in cells.

Effects of free amino acids on RNA folding and catalysis in vitro are generally considered to be inhibitory. For instance, discoveries of amino acid-sensing RNAs including the lysine riboswitch, arginine riboswitch, and glutamine riboswitch revealed that a simple amino acid can bind tightly to the specific loops in the riboswitches and change local RNA structures[14–16]. Furthermore, self-splicing of the group I intron is inhibited by arginine;[17,18] myriad RNA secondary and tertiary structures are destabilized in proline;[19] and arginine, lysine, and betaine—an analog of glycine—destabilize tertiary interactions in a group I intron[20]. In contrast, general effects of amino acids on RNA function under in cellular conditions are largely unknown.

Herein, we investigate the catalysis and folding of several functional RNAs in the presence of amino acid-chelated magnesium under cellular conditions. We test activity of HDV-like, hammerhead, and *glmS* self-cleaving ribozymes, which are widely found in the three domains of life, with amino acid-chelated magnesium[21–26]. The HDV-like ribozyme is one of the best studied ribozymes: it folds into a pseudoknot structure through two co-axial stacks and employs general acid-base catalysis for the self-cleavage reaction utilizing metal ion and nucleobase catalysis (Supplementary Fig. 2). Binding of magnesium is essential not only for RNA catalysis in many ribozymes but also for RNA folding. An NMR study and $Tb^{3+}$-induced cleavage assay suggests up to eight magnesium ions could bind with the CPEB3 ribozyme and several of these sites are important for folding[27]. In addition, in the hammerhead and *glmS* ribozymes, $Mg^{2+}$ serves folding and catalysis roles[24,25].

To analyze the effects of metal ions on RNA folding, stability, and catalysis under cellular conditions, we focus on interactions between the most prevalent divalent ion and metabolites, magnesium ion and amino acids, respectively. Every amino acid has one carboxylate group, which is deprotonated at cellular pH and enables association with $Mg^{2+}$. Although the affinity of amino acids for magnesium is relatively weak (in the millimolar range), this interaction is still detectable by titration and NMR experiments in aqueous systems and accounts for ~15 mM of all chelated $Mg^{2+}$ (see Results)[28–30]. We asked whether such weakly chelated magnesium can affect RNA folding and catalysis. Herein, we report that weakly bound magnesium strongly promotes RNA folding, stability, and catalysis, which affect nearly every biological process involving RNA.

## Results

**Amino acid-chelated magnesium in *E. coli* cells.** To test whether RNA function is affected by amino acid-chelated magnesium, we first estimated its concentration in *E. coli* cells. Total concentrations of metabolites and their binding affinities for magnesium have been previously reported (Supplementary Fig. 1 and Supplementary Table 1)[13,28,31]. Our estimation based on chemical equilibrium (see Methods) shows that when the total concentration of magnesium in *E. coli* cells is 54 mM, ~14 mM $Mg^{2+}$ is associated with the four most abundant amino acids in living cells (aa$_4$CM). Among amino acids, the concentration of glutamate is highest in *E. coli* at 96 mM, and despite having a relatively weak $K_D$ of 15 mM, it binds 11 mM $Mg^{2+}$ in the cell (Supplementary Table 1).

**Amino acid-chelated magnesium promotes RNA folding.** We began our studies by testing whether amino acid-chelated magnesium can promote RNA thermostability. We first measured RNA thermal denaturation in 0, 9.6, and 96 mM glutamate in the absence of magnesium using the HDV-like ribozyme drz-spur-3, which melts in a single transition (Supplementary Fig. 3a and Supplementary Table 2). Upon addition of 96 mM potassium glutamate (total potassium ion is 236 mM), the melting transition peak shifted slightly higher (~2 °C) and the enthalpy ($\Delta H°$) decreased slightly (approximatley −8 kcal·mol$^{-1}$). We hypothesize that this small effect is derived from the cation, since additional 96 mM potassium chloride (total potassium ion is also 236 mM) stabilized the RNA to approximately the same small extent (Supplementary Fig. 3b). The relative change in $\Delta\Delta G$ in the presence of 96 mM glutamate or additional KCl in 0 mM $Mg^{2+}$ are modest and similar at −1.9 ± 0.1 and −2.6 ± 0.2, respectively (Supplementary Table 2).

Next, we evaluated effects of weak $Mg^{2+}$ binding via glutamate-chelated magnesium (gluCM) and aa$_4$CM in 2, 0.5 and 0.01 mM $Mg^{2+}_{free}$ conditions (Fig. 1, Supplementary Table 2). As a control, we used EDTA-chelated magnesium (EDTACM) since EDTA binds magnesium relatively tightly[32]. Melting curves in 2 mM $Mg^{2+}_{free}$ in the presence of gluCM, aa$_4$CM, or EDTACM are provided in Fig. 1. GluCM and aa$_4$CM, which have 11.3 mM $Mg^{2+}_{complex}$/2 mM $Mg^{2+}_{free}$ and 14 mM $Mg^{2+}_{complex}$/2 mM $Mg^{2+}_{free}$, respectively, clearly increase RNA stabilization (~5 °C), and the degree of the stabilization was comparable to that with 13.3 mM $Mg^{2+}_{free}$ (Fig. 1a, compare red and cyan to orange). In contrast, stabilization was not observed in the presence of EDTACM, which also had 11.3 mM $Mg^{2+}_{complex}$/2 mM $Mg^{2+}_{free}$ (Fig. 1a, compare green to blue). Clearly, RNA stabilization is greater when affinity of the small molecule chelator for $Mg^{2+}$ is weak. Similar results were obtained in 0.5 and 0.01 mM $Mg^{2+}_{free}$ (Fig. 1b and c). Thus, weakly chelated magnesium contributes strongly to RNA thermostability.

**Amino acids protect RNA from magnesium-dependent degradation.** Pioneering studies from Adamala and Szostak demonstrated that citrate-chelated $Mg^{2+}$ can protect single-stranded (ss) RNA from degradation[33]. In that report, RNA degradation experiments were conducted in the presence of 200 mM citrate and 50 mM magnesium. The absolute abundance of citrate, however, is just 2.0 mM in *E. coli* cells[12,13], or ~100-fold less than in those studies. We wondered whether aa$_4$CM can reduce magnesium-catalyzed degradation of RNAs, including RNAs with secondary and tertiary structure. To test this, we performed successive melting experiments using drz-spur-3 ribozyme (Fig. 2). First, experiments were conducted in the absence of chelator. In the case of 13.3 mM $Mg^{2+}_{total}$, the

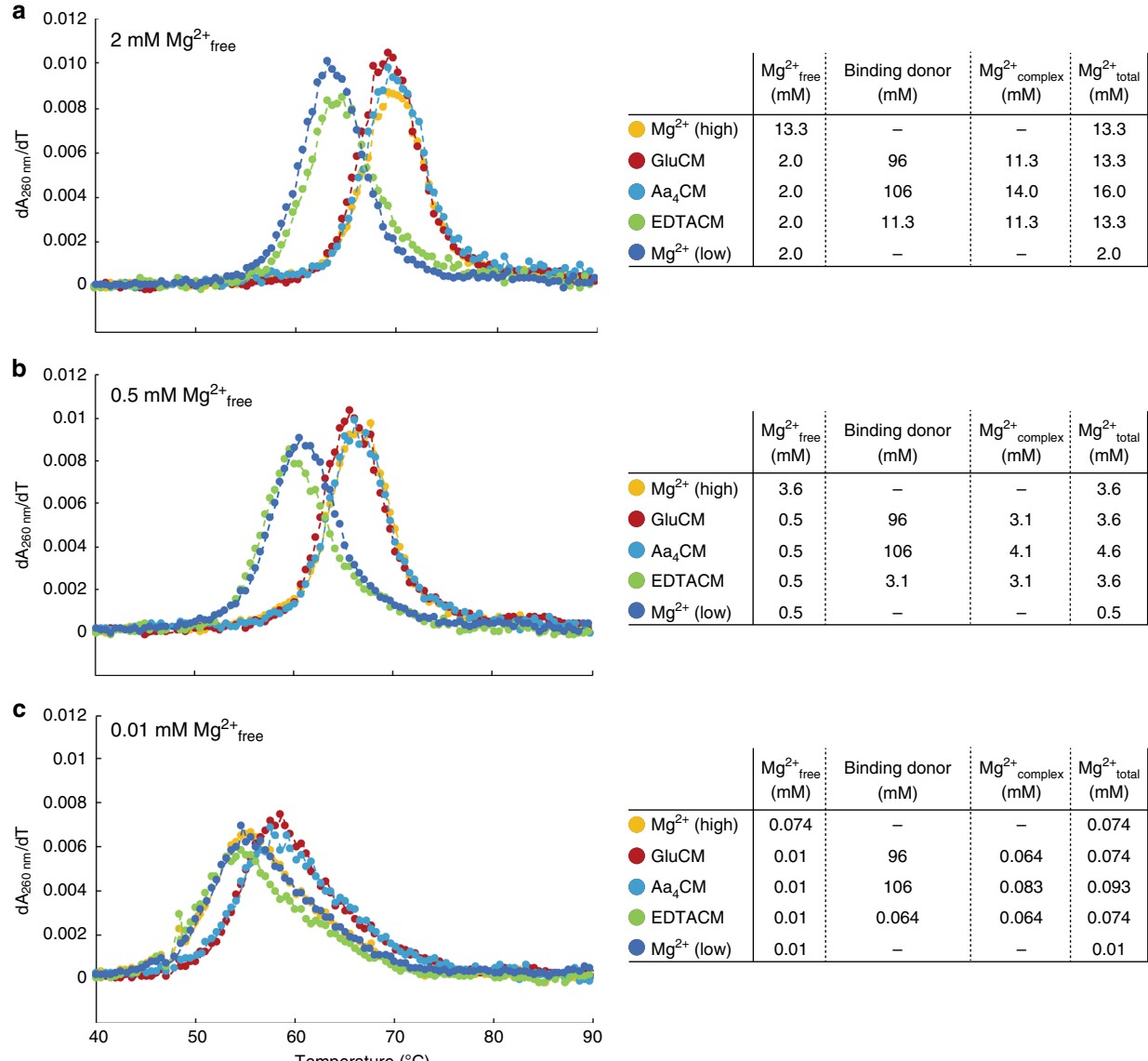

**Fig. 1** Amino acid-chelated magnesium promotes RNA thermostability. **a** Melting curves of drz-spur-3 ribozyme in bacterial conditions; 13.3 mM $Mg^{2+}_{total}$ (orange: $Mg^{2+}$ high), 96 mM glutamate and 13.3 mM $Mg^{2+}_{total}$ (red: GluCM), 106 mM amino acids and 16.0 mM $Mg^{2+}_{total}$ (cyan: $Aa_4CM$), 11.3 mM EDTA and 13.3 mM $Mg^{2+}_{total}$ (green: EDTACM), and 2 mM $Mg^{2+}_{total}$ (blue: $Mg^{2+}$ low). **b**, **c** Melting curves in **b** eukaryotic conditions and **c** 0.01 mM $Mg^{2+}_{free}$ condition, chosen to test $Mg^{2+}$ sensitivity. The same colors as **a** indicate the same concentrations of the binding donors with different total magnesium concentrations (3.6 mM, 4.6 mM, or 0.5 mM, and 0.074 mM, 0.093 mM, or 0.01 mM, respectively)

maximum $dA_{260\ nm}\cdot dT^{-1}$ values decreased to near 0 in the second thermal denaturation owing to RNA degradation, while in the case of 0.074 mM $Mg^{2+}_{total}$, the maximum $dA_{260\ nm}\cdot dT^{-1}$ value decreased to ~60% of the maximum $dA_{260\ nm}\cdot dT^{-1}$ value in the first melting curves. Next, we conducted experiments in the presence of either gluCM, aa₄CM or EDTACM. The unfolding transition peak for the second melting curves in the presence of gluCM was observed even in 13.3 mM $Mg^{2+}_{total}$/2 mM $Mg^{2+}_{free}$, where gluCM decreased degradation by ~35% (Fig. 2). In aa₄CM, RNA degradation was inhibited by ~70%, similar to that in EDTACM of ~80%. Similar effects were found in 3.6 mM $Mg^{2+}_{total}$/0.5 mM $Mg^{2+}_{free}$ and 0.074 mM $Mg^{2+}_{total}$/0.01 mM $Mg^{2+}_{free}$ conditions. Notably, in 0.5 mM $Mg^{2+}_{free}$, RNA degradation was reduced by ~60% and ~80% in gluCM and aa₄CM, respectively. Presumably aa₄CM is better at preventing degradation because some of these amino acids bind $Mg^{2+}$ tighter than glutamate. In

sum, gluCM and aa₄CM significantly prevent degradation of a functional RNA with both secondary and tertiary structure at high-temperature in physiological conditions.

Next, we tested for protection from RNA degradation at constant temperature, choosing the physiological temperature of 37 °C in 2 mM $Mg^{2+}_{free}$ using drz-spur-3. We perform in-line probing (ILP) experiments, which reveal RNA degradation at the nucleotide level (Fig. 3)[34]. By comparison of ILP reactivities between 13.3 mM $Mg^{2+}_{free}$ and gluCM conditions of 13.3 mM $Mg^{2+}_{total}$/2 mM $Mg^{2+}_{free}$, although the pattern of degradation is not dramatically changed (Fig. 3a; lanes 4–9) the intensity of each band is reduced in the gluCM condition by 20–40% (Fig. 3b). Thus, gluCM reduces $Mg^{2+}$-promoted in-line degradation. In aa₄CM conditions, RNA degradation reduced even more, by 40–70% (Fig. 3a; lanes 10-12). Thus, weakly chelated magnesium significantly prevents RNA degradation at physiological

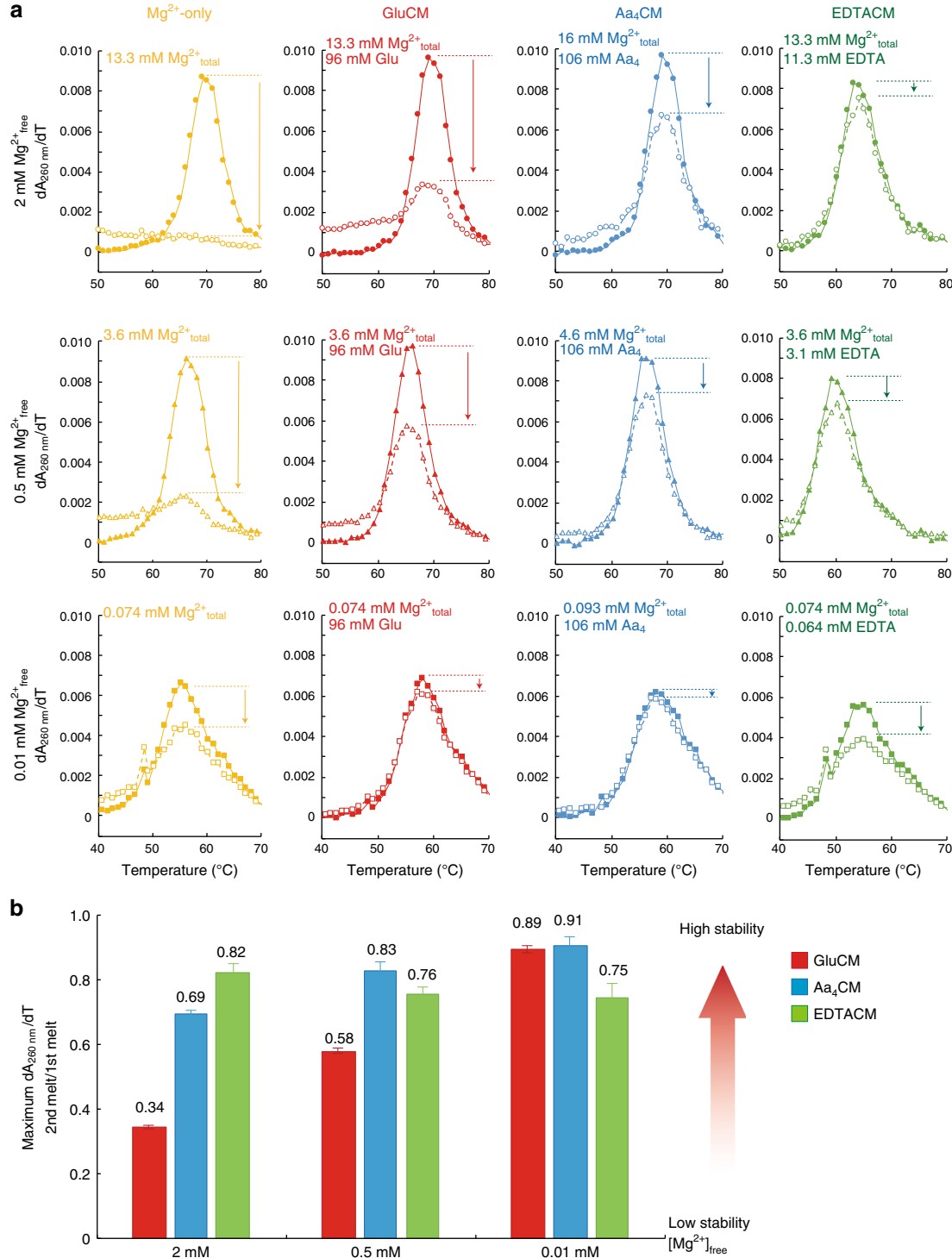

**Fig. 2** Amino acid-chelated magnesium reduces RNA degradation according to thermal denaturation experiment. Successive thermal denaturation experiments are conducted using cleaved drz-spur-3 ribozyme. **a** Closed and open symbols indicate first and second successive thermal denaturation curves, respectively. Arrows indicate decrease of the maximum $dA_{260nm} \cdot dT^{-1}$ from the first melting curves to the second melting curves. Free $Mg^{2+}$ is 2 mM, 0.5 mM, or 0.01 mM, while chelated $Mg^{2+}$ was also present and achieved in different chelators as provided in the figure. **b** Summary of RNA stabilization in each condition. RNA stabilization at a given condition was estimated by dividing the maximum $dA_{260 nm} \cdot dT^{-1}$ in the second melting curves by that in the first melting curves

temperatures. Extra reduction compared to 2 mM free $Mg^{2+}$ was observed in the presence of EDTA and 2 mM $Mg^{2+}_{free}$ (Fig 3a; lanes 13-15), which may be due to weak chelation of additional $Mg^{2+}$ ions by $[EDTA-Mg^{2+}]^{2-}$ complexes. Figure 3c confirms that the major sites of ILP (blue circles) were near loops as expected, and reveals that gluCM offered protection from

degradation in helical regions (P2, P3, P4), as well as an unstructured loop region (L3) (red arrows)[35].

**Amino acid-chelated magnesium enhances RNA catalysis.** Next, we performed self-cleaving assays beginning with the small

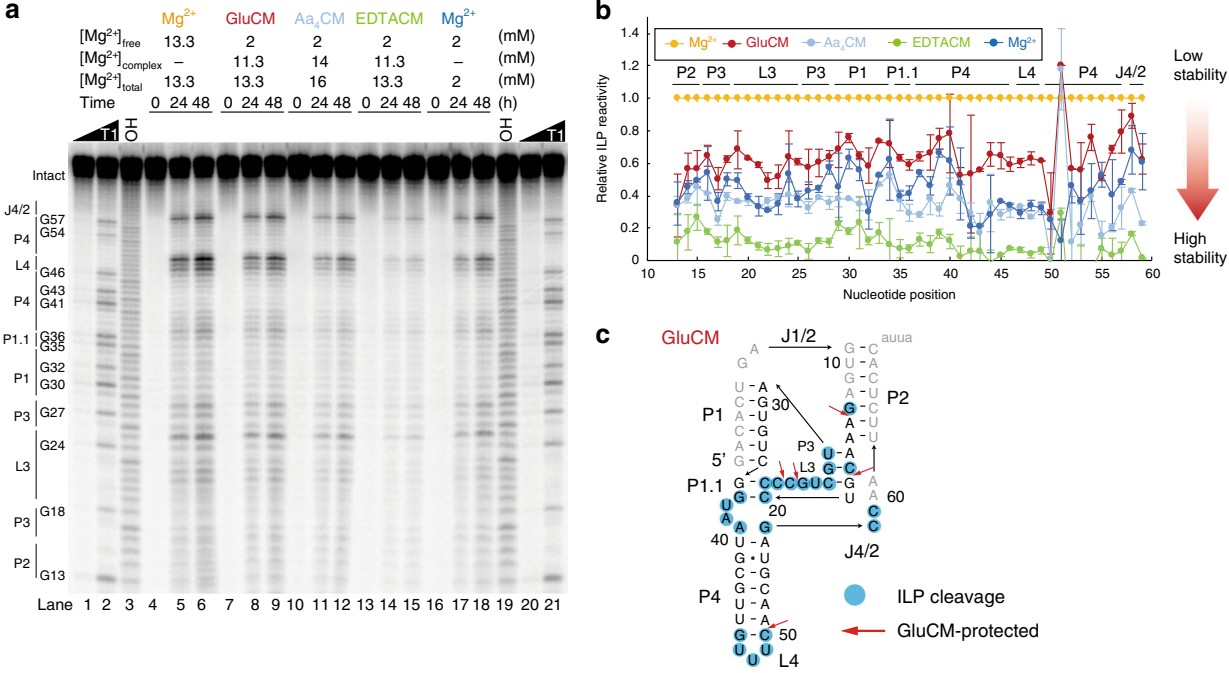

**Fig. 3** Amino acid-chelated magnesium reduces RNA degradation according to ILP reactivities. **a** ILP on the drz-spur-3 ribozyme. Lane 1–2 and 20–21 are RNase T1 ladders (G). Lane 3 and 19 are alkaline hydrolysis ladders (OH⁻). Lane 4–6 are time-dependent ILPs in the presence of 13.3 mM $Mg^{2+}_{total}$. Lane 7–9 are that in the presence of 96 mM glutamate and 13.3 mM $Mg^{2+}_{total}$ (2 mM $Mg^{2+}_{free}$). Lane 10–12 are that in the presence of 106 mM amino acids and 16 mM $Mg^{2+}_{total}$ (2 mM $Mg^{2+}_{free}$). Lane 13–15 are that in the presence of 11.3 mM EDTA and 13.3 mM $Mg^{2+}_{total}$ (2 mM $Mg^{2+}_{free}$). Lane 16–18 are that in the presence of 2 mM $Mg^{2+}_{total}$ for a control. Concentrations of binding donors are the same as in Fig. 1. **b** Relative ILP reactivity at 24 h at each nucleotide position is shown. The relative reactivity in high $Mg^{2+}$ condition was set at 1.0. The error bars show average errors ($n = 2$). **c** ILP reactivity in the presence of 13.3 mM $Mg^{2+}$ referenced to U49. Nucleotides colored in blue show the positions where ILP reactivity is greater than 7.5% of U49. Red arrows denote positions of ILP reactivity where gluCM offers protection

ribozyme CPEB3 ribozyme (Fig. 4). In the presence of 0.1 mM $Mg^{2+}_{free}$ and 0.5 mM $Mg^{2+}_{free}$, with gluCM of 0.66 and 3.1 mM, respectively, self-cleavage was unmistakably accelerated, by up to 17-fold and 2.6-fold, respectively. These values were similar to fold-stimulations for when total concentrations of $Mg^{2+}$ in the presence of Glu were allowed to be all-free (Figs. 4a, 4b and Supplementary Table 3). In contrast, in the presence of 3.1 mM EDTACM in 0.5 mM $Mg^{2+}_{free}$, the rate was almost same as 0.5 mM $Mg^{2+}_{free}$ condition (Fig. 4b) indicating that chelation must be weak to have an effect on catalysis. In the presence of 2 mM $Mg^{2+}_{free}$, gluCM of 11.3 mM had only a slight, 1.6-fold stimulation on rate (Fig. 4c and Supplementary Table 3). A small rate enhancement at 2 mM $Mg^{2+}_{free}$ suggests that the free magnesium concentration might already be near saturation in the CPEB3 ribozyme reaction (Fig. 4c). In addition, the extent of reaction was greater in the presence of weakly chelated $Mg^{2+}$ in 0.1 and 0.5 mM $Mg^{2+}_{free}$, but not in 2 mM $Mg^{2+}_{free}$.

To test whether this finding extends to other ribozymes, we assayed self-cleavage of the *glmS* ribozyme and hammerhead ribozyme 16 (HH16) under identical conditions (Supplementary Fig. 4 and Supplementary Fig. 5). In the case of *glmS*, for 0.5 mM $Mg^{2+}_{free}$, the 3.1 mM gluCM stimulated the reaction by almost 5-orders of magnitude. In 0.1 mM $Mg^{2+}_{free}$, chelated $Mg^{2+}$ also greatly stimulated cleavage, although fold stimulation could not be calculated because the background reaction was barely detectable. In 2 mM $Mg^{2+}_{free}$, the gluCM of 11.3 mM had a significant albeit smaller 2.8-fold stimulation. (Supplementary Table 4). For the hammerhead ribozyme, rate stimulations were 17- and 3.6-fold in the same gluCM in 0.5 mM $Mg^{2+}_{free}$ and 2 mM $Mg^{2+}_{free}$, respectively (Supplementary Table 5). Also, we saw that the amplitude of $f_{cleaved}$ is dependent on RNA species and $Mg^{2+}_{free}$ concentration. For example, the amplitudes of CPEB3 and

*glmS* ribozyme reactions in the presence of gluCM in 0.1 mM $Mg^{2+}$ are increased by 1.7-fold and >10-fold, respectively. This is probably because the concentration of magnesium that is minimally required to fold into the catalytic forms for the ribozymes is RNA-species dependent.

Taken together, we conclude that weakly chelated magnesium can drastically improve RNA catalysis in the conditions that mimic the cellular environment.

**Amino acid-chelated magnesium promotes RNA compaction.** To clarify the mechanism of enhancement of RNA catalysis by amino acid-chelated magnesium, we measured RNA folding using small-angle X-ray scattering (SAXS) (Supplementary Fig. 6 and Supplementary Table 6). We collected SAXS data in four conditions (10 mM $Mg^{2+}_{free}$, 0.5 mM $Mg^{2+}_{free}$, 9.5 mM gluCM, and 9.5 mM EDTACM) using the CPEB3 ribozyme. In the EDTACM condition (9.5 mM EDTACM/0.5 mM $Mg^{2+}_{free}$), we saw inter-particle repulsion that was not observed in the comparative condition of 0.5 mM $Mg^{2+}_{free}$, which implies that the EDTA chelates residual magnesium and non-specifically binds a second magnesium, which is consistent with the ILP experiments (Fig. 3), and the apparent concentration of magnesium in the solution is slightly less than 0.5 mM $Mg^{2+}_{free}$. In such a condition, the RNA phosphate charge is less neutralized. Herein we show the other SAXS data. Dimensionless Kratky and $p(r)$ plots are provided in Supplementary Fig. 6a and 6b, where the ribozyme in each condition was shown to be well-folded and to be a monomer structure in the solution, respectively. The collected scattering data were evaluated by overlaying with theoretical scattering data created from the modeled crystal structure for the CPEB3 ribozyme[36], where the experimental data are similarly

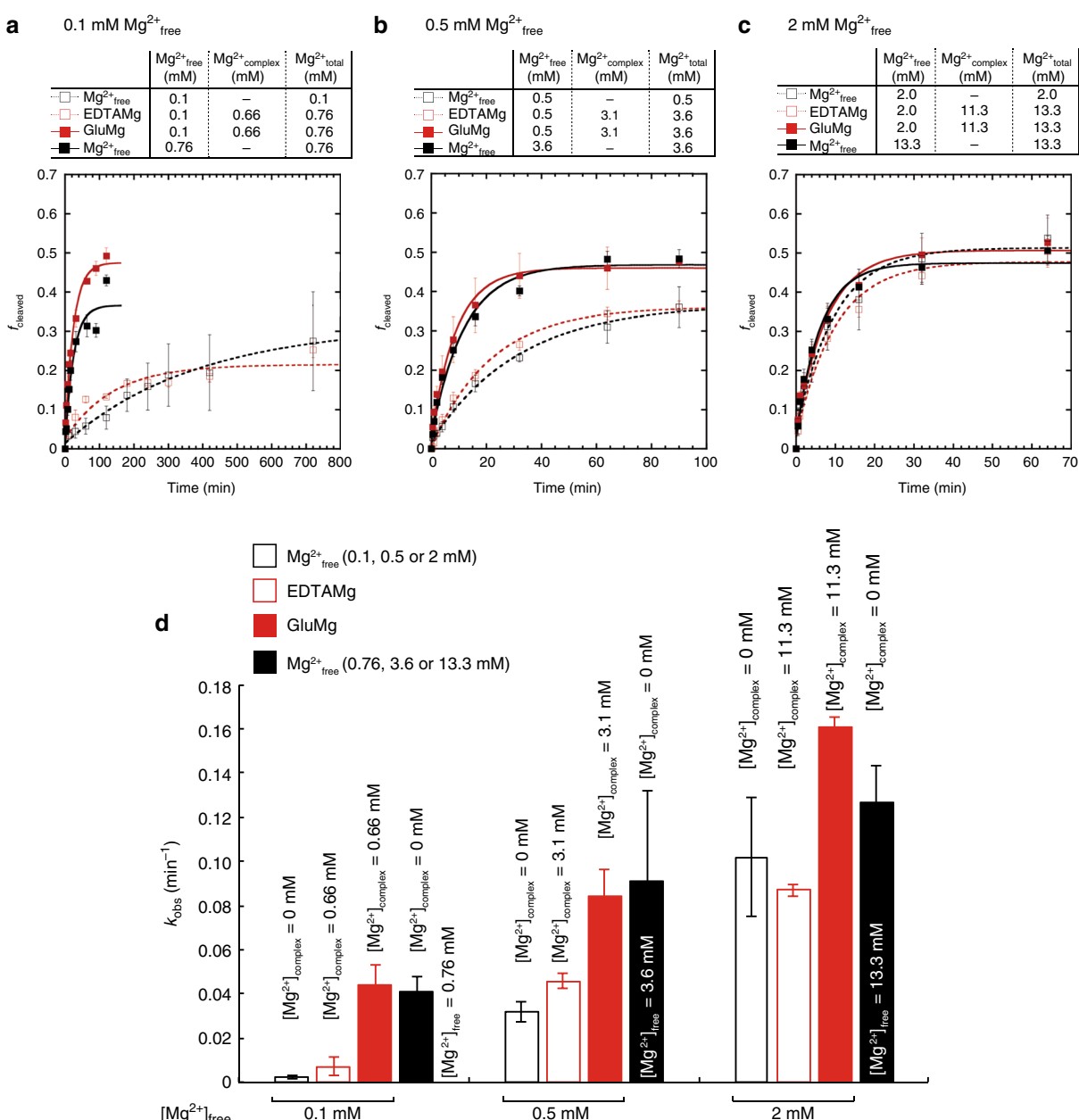

**Fig. 4** Amino acid-chelated magnesium stimulates self-cleavage of the CPEB3 ribozyme. **a** Self-cleaving reaction at 0.1 mM $Mg^{2+}_{free}$ condition. Fraction ($f_{cleaved}$) vs time plot is shown. Black open squares show 0.1 mM $Mg^{2+}_{total}$ as a control, red open squares show 0.66 mM EDTA, red filled squares show 96 mM glutamate, and black squares show 0.76 mM $Mg^{2+}_{total}$. **b** Self-cleaving reaction at 0.5 mM $Mg^{2+}_{free}$ condition. **c** Self-cleaving reaction at 2 mM $Mg^{2+}_{free}$ condition. Concentrations of binding donors for **b** and **c** are provided in Fig. 1; for **a**, binding donor concentrations are 96 mM and 0.66 mM for glutamate and EDTA. **d** $k_{obs}$ values for the cleaving reaction at the various magnesium conditions. The error bars mean S.D. ($n = 4$). The symbols and colors in **b**, **c**, and **d** are same with **a**

fitted to the theoretical model (Supplementary Fig. 6c, 6d, and 6e). Radius of gyration ($R_g$) increases from 25.7 Å in 10 mM $Mg^{2+}_{free}$ to 27.0 Å in 0.5 mM $Mg^{2+}_{free}$, whereas $D_{max}$ also increases from 71 Å in 10 mM $Mg^{2+}_{free}$ to 75 Å in 0.5 mM $Mg^{2+}_{free}$. $R_g$ and $D_{max}$ in 9.5 mM gluCM are 26.3 Å and 72 Å, respectively, showing more of a compact structure for the CPEB3 ribozyme in the gluCM condition compared to that in the 0.5 mM $Mg^{2+}_{free}$ condition. We next created bead models from the SAXS data and overlaid them on the modeled structure for the CPEB3 ribozyme (Supplementary Fig. 6f, 6g, and 6h). These bead models support the notion that gluCM contributes to compaction of the RNA (compare Supplementary Fig. 6g and 6h).

Next, we tested the effects of aaCM directly on RNA catalysis. We chose the *glmS* ribozyme for these experiments because this ribozyme has been extensively investigated by our group and provides us a good benchmark to compare[25]. We carried out sulfur substitution at the non-bridging oxygens of the scissile phosphate to test whether the mechanism changes in the presence of aaCM (Supplementary Fig. 7 and Supplementary Table 7). If amino acid-chelated magnesium can participate in *glmS* ribozyme catalysis directly, a thio effect, which is calculated from rate constants for oxo, $R_p$, $S_p$, and dithio substrates (Supplementary Fig. 7a) as defined in the Methods, would change in the presence or absence of amino acid-chelated magnesium. We first

investigated thio effects with unchelated $Mg^{2+}$. We found that the thio effect in 10 mM $Mg^{2+}_{free}$ (measured previously) and 3.6 mM $Mg^{2+}_{free}$ (measured here) were similar (200 and 230 for the $R_p$ substrate, 15 and 30 for the $S_p$ substrate, and 9000 and 25,700 for the dithio substrate, respectively)[25], whereas essentially no thio effect was found in 0.5 mM $Mg^{2+}_{free}$, presumably because of a higher potential barrier for the chemical reaction and/or folding due to the low magnesium concentration (Supplementary Fig. 7b, 7c and 7d). Next, we measured the thio effects in the presence aaCM. The thio effect for each substrate in 0.5 mM $Mg^{2+}_{free}$ with 3.1 mM gluCM was similar to that in 3.6 mM $Mg^{2+}_{free}$ (Supplementary Fig. 7c and 7e). Similarity of these values suggests that gluCM is not directly involved in the reaction mechanism. Presence of thio effects in 0.5 mM $Mg^{2+}_{free}$ only in the background of 3.1 mM gluCM suggests that 3.1 mM gluCM affect RNA folding states.

To provide more insight into the mechanism of the effect of aaCM on RNA catalysis, we also analyzed free activation energy ($E_a$) for the CPEB3 ribozyme reaction (Supplementary Fig. 8 and Supplementary Table 3). If aaCM promotes the ribozyme activity not by participating in catalysis, the $E_a$ value in aaCM condition should be comparable to that in magnesium condition. Consistent with this notion, the slopes in both conditions are nearly parallel, and calculated $E_a$ values in 3.1 mM gluCM/0.5 mM $Mg^{2+}_{free}$ is 16.7 kcal·mol$^{-1}$, whereas the $E_a$ values in 0.5 mM $Mg^{2+}_{free}$ is 19.4 kcal·mol$^{-1}$, which is a similar value (Supplementary Fig. 8). Overall, our results support the model that aaCM promotes ribozyme activity not by directly participating in catalysis by either the metal or amino acid, but rather by indirectly promoting compaction of the ribozyme, which assembles the active site.

**Crowding condition increases fraction of active ribozyme**. Previous work revealed that the molecular crowding agents increase ribozyme activity and stability through RNA compaction[4,8,36,37]. We asked whether RNA catalysis is synergistically accelerated in more physiological condition where we tested the CPEB3 activity in the presence of molecular crowding agent (20% PEG8000) with aaCM (Supplementary Fig 9a-9c). A synergistic effect of chelated magnesium and 20% PEG8000 was not observed on rate constants, as these were not significantly changed in the presence and absence of 20% PEG8000 (Supplementary Fig 9d-9f and Supplementary Table 3). However, the amplitudes of $f_{cleaved}$ in the presence of the crowder in both 0.1 mM gluCM and 0.1 mM EDTACM increased by up to 2-fold (Supplementary Fig 9g), which is consistent with the previous report[8,36]. This effect arises from increase of fraction of active ribozyme, consistent with our SAXS experiments (Supplementary Fig. 6).

**Other metabolite-chelated magnesium also increases RNA stability**. Based on our observations, we wondered if other metabolites that can bind magnesium could show the similar effect on RNA folding as aaCM. To address this, we performed thermal denaturation experiments in the background of 3 mM $Mg^{2+}_{total}$, combined with 28 mM chelator including malate and citrate with $K_D$s of 19 and 0.4 mM, respectively (Fig. 5a)[31]. The thermal denaturation curve in the presence of 1.9 mM of gluCM and 1.1 mM $Mg^{2+}_{free}$ is identical with that in the presence of 3 mM $Mg^{2+}_{free}$ condition (Fig. 5a, compare red to orange), in agreement with Fig. 1. The stabilization of RNA by 1.7 mM of malate-bound magnesium in 1.3 mM $Mg^{2+}_{free}$ is less than that by gluCM despite having slightly more free $Mg^{2+}$ (Fig. 5a, compare green to red). As expected, citrate- and EDTA-chelated magnesium, which have only 0.05 mM and ~0 mM $Mg^{2+}_{free}$, did not

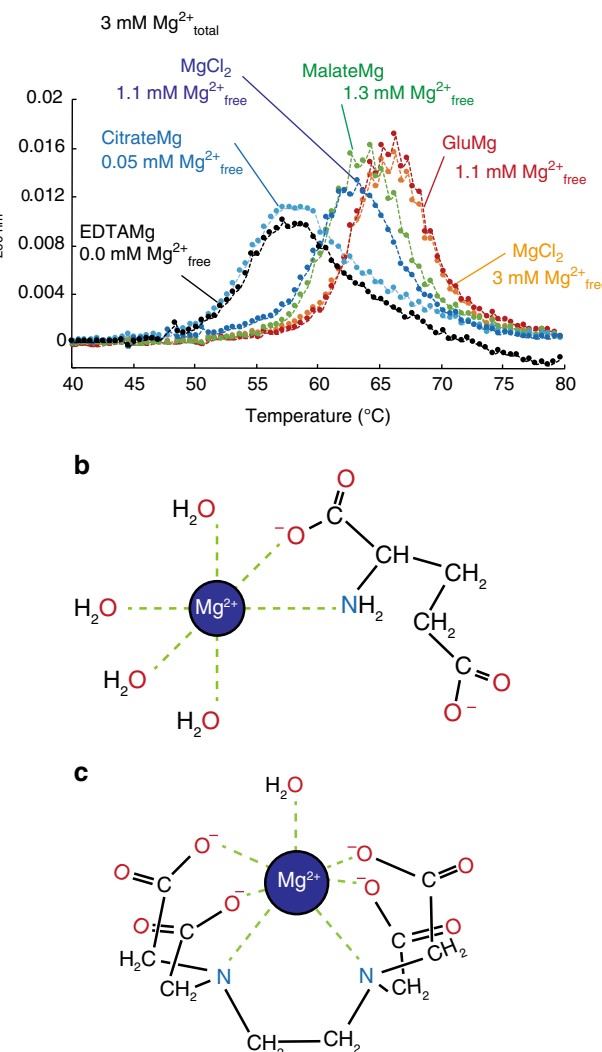

**Fig. 5** Coordinated water molecules are critical for RNA stabilization. **a** Melting curves of drz-spur-3 ribozyme in the background of metabolites. 3 mM $Mg^{2+}_{total}$ (orange) and 28 mM of glutamate (red), malate (green), citrate (cyan), or EDTA (black) conditions were tested. As a control, 1.1 mM $Mg^{2+}_{total}$ (blue) condition was performed. The estimated concentrations of $Mg^{2+}_{free}$ are provided. $K_D$'s are given in the main text. **b** Chemical structure of glutamate and magnesium. Glutamate interacts with magnesium through N, O-chelation. **c** Chemical structure of EDTA and magnesium. EDTA interacts with magnesium through O, O, O, O, N, N-chelation

promote RNA stability to the same extent as 1.1 mM $Mg^{2+}_{free}$ (Fig. 5a compares cyan and black). GluCM and malate-chelated $Mg^{2+}$ have ~4 chelated waters[38]. Thus, we found that the dissociation constant for magnesium and number of coordinated water molecules have positive correlation to RNA stabilization.

## Discussion

In this study, we found that physiological concentrations of amino acid-chelated magnesium promote RNA thermostability, protect RNA from magnesium-dependent degradation in both heating and in constant physiological temperatures, and greatly enhance the reaction of self-cleaving ribozymes through RNA compaction. Our finding that aaCM promotes catalysis indirectly by compacting the RNA rather than directly by participating in catalysis supports a general effect, consistent with our observed

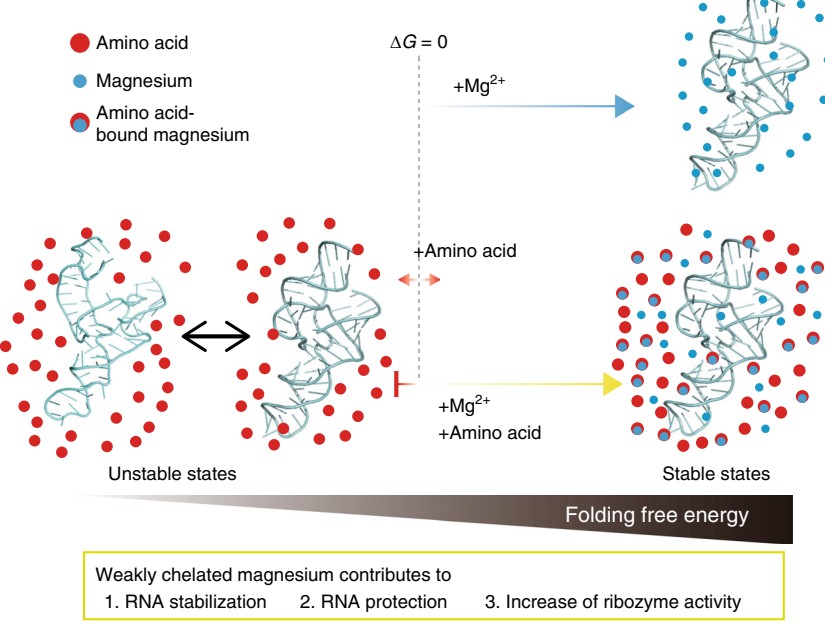

**Fig. 6** Proposed effects of weakly chelated magnesium on RNA. Illustration of the effects of amino acids and aaCM for RNA. Amino acids alone increase the folding free energy of RNA, either having no effect or destabilizing the RNA. At the same time, aaCM decreases the folding free energy of RNA, stabilizing the RNA and promoting high catalytic activity

promotion of the activity of three different ribozymes. Nakano and co-workers reported that anionic compounds, including glutamate, reduce the rate of hammerhead ribozyme activity by decreasing the amount of free $Mg^{2+}$[39]. However, in their studies the concentration of free $Mg^{2+}$ was not held constant as it is in the cell; moreover, addition of 32.5 mM glutamate in the background of 5 mM $Mg^{2+}$ total reduced hammerhead ribozyme activity by a negligible amount, ~1.15 ± 0.07-fold[39], consistent with our findings.

Our data suggest that gluCM has apparently no effect on secondary and tertiary structures of the RNAs. Similar cleavage patterns are observed in each condition in the ILP experiments, although cleavage intensity is changed in the presence of gluCM (Figs. 3a, b), indicating similar secondary structure of the drz-spur-3 ribozyme in each condition. Bead models of the CPEB3 ribozyme calculated from SAXS data are well-fitted in the crystal structure and show similar overall structures of the CPEB3 ribozyme in the presence and absence of gluCM (Supplementary Fig. 6f-6h). To quantify the extent of thermostability, we calculated apparent relative changes in free energy ($\Delta\Delta G$ values) with respect to the control experiments of no chelated $Mg^{2+}$ (Supplementary Table 2) and found that amino acid-chelated $Mg^{2+}$ decrease $\Delta\Delta G$ values significantly. For instance, $\Delta\Delta G$ in the presence of 11.3 mM gluCM in 2 mM $Mg^{2+}_{free}$ condition is –6.9 ± 0.3 kcal · mol$^{-1}$, whereas 11.3 mM EDTACM did not change the values (e.g. $\Delta\Delta G$ in the presence of EDTACM in 2 mM $Mg^{2+}_{free}$ condition is + 0.4 ± 0.5 kcal · mol$^{-1}$). Folding free energy in the presence of aa$_4$CM is also ~7 kcal· mol$^{-1}$ more favorable than without chelated $Mg^{2+}$, indicating that aa$_4$CM contributes strongly to RNA folding. This is depicted in Fig. 6, where amino acids generally destabilize the RNA, $Mg^{2+}$ stabilizes the RNA, and aa$_4$CM stabilizes the RNA to a similar amount. Given that the aa$_4$CM allowed the RNA to fold cooperatively (i.e., in a single all-or-none transition), the aa$_4$CM may promote tertiary stability.

Crystal structures of gluCM and EDTACM are provided in Fig. 5[32,38]. In gluCM, chelated $Mg^{2+}$ is hexacoordinated, with one oxygen of the main chain carboxyl group and the amino group of glutamate and four $H_2O$ molecules in the inner-coordination

sphere in basic conditions (Fig. 5b)[38]. Interestingly, the chelated moieties are all from the main chain of glutamate (carbonyl and amine) and the side chain carboxyl group is not chelating, suggesting that perhaps all amino acids can chelate with $Mg^{2+}$ similarly in cells. In contrast, the EDTACM is seven-coordinated, with four carboxyl groups and two nitrogen atoms of EDTA and one water molecule (Fig. 5c). Based on these crystal structures, we suggest a positive relationship between RNA folding and catalysis and the number of coordinated water molecules chelated to $Mg^{2+}$.

Previous work on RNA behavior in the presence of crowding agents revealed that the crowding environment plays an important role in assisting RNA folding without specific interaction between the crowding agents and RNA: it decreases folding heterogeneity and increases folding cooperatively[7,40]. This arises by destabilizing secondary structure and stabilizing tertiary structure of RNAs[4,9,19,40]. Furthermore, crowding agents increase RNA catalysis[8,36]. The crowding environment decreases the threshold binding constant of $Mg^{2+}$ ~7-fold that is required to RNA folding into active forms, and the apparent binding constant is millimolar order[7]. In such cellular mimic condition, the weakly chelated $Mg^{2+}$ may further increase RNA folding and catalysis by a synergistic effect. Our results show that fraction of active ribozyme is certainly increased by the crowding environment, which is consistent with our previous report[36], whereas there is no stimulation on rate constant (Supplementary Fig. 9). In other words, the chelated magnesium in crowding solution has a further positive effect on RNA folding but no effect on RNA catalysis suggesting that the chelated magnesium behaves as unbound magnesium. Two possible interpretations of the effects of weakly cheated $Mg^{2+}$ are considered: binding competition of $Mg^{2+}$ between amino acid and RNA, or sharing the ions. Magnesium ions have several modes for interaction with RNA including diffuse, outersphere, and innersphere binding, which have decreasing levels of solvation[41,42]. Both diffuse and outer-sphere ions have water molecules fully coordinated to them, while innersphere ions typically have several water molecules coordinated to them. In some cases, the RNA may directly compete with

the weak chelator for the $Mg^{2+}$ and scavenge the ion; estimated $K_D$s for binding of $Mg^{2+}$ to specific sites on RNA are in the micromolar to millimolar range[41], suggesting that such competition is possible. In other cases, the chelator-bound $Mg^{2+}$ ion may interact with the RNA, in effect "sharing" the $Mg^{2+}$ ion with the RNA; observations that weak chelator reduced hydrolysis in our ILP studies and decreased $R_g$ and $D_{max}$ in our SAXS data support contribution from this mode of ion binding. Future studies may be directed at identifying the relative contributions of these and other modes of ion binding to RNA function. Theoretical studies may provide insight into the generality of the findings herein by estimating which roles aa$_4$CM can perform and estimating the dynamics of functional RNAs containing amino acid-chelated $Mg^{2+}$. It is further possible to imagine aa$_4$CM delivering the functionality of imidazole or other amino acids to the active site of a ribozyme, thereby enhancing its functionality in proton transfer, charge stabilization, or hydrogen bonding.

The work conducted herein has further implications for origin of life chemistry. Adamala and Szostak demonstrated that chelation of $Mg^{2+}$ by citrate promotes formation of fatty acid membranes and protects ssRNA and short duplex RNAs from degradation[33]. In addition, Keating and colleagues showed that intermediate chelation of $Mg^{2+}$ is critical for formation of liposome-stabilized aqueous two-phase emulsions for application in ribozyme-containing bioreactors[43]. The intermediate chelation also plays significant role in biomimetic mineralization by preventing $Ca^{2+}$-mediated aggregation of lipid vesicles[44]. The work reported herein shows that amino acids can provide weak divalent ion chelation, possibly providing a bridge between the RNA world and the RNP world, and even supporting their co-evolution[45]. We find that not only ssRNA and short duplex RNA are protected from degradation by glutamate, but that complex RNAs like ribozymes are also protected; moreover, the stability and catalytic ability of ribozymes are promoted. One possibility is that glutamate chelation prevents metal ions from interacting with the 2′OH, although it is possible that the solvation pattern around the RNA is altered by the glutamate. While citrate provided stronger protection of ssRNA to $Mg^{2+}$-promoted degradation (~85% reduction of the degradation as compared with no citrate condition[33]), citrate-chelated $Mg^{2+}$ did not improve thermostability of the CPEB3 ribozyme (Fig. 5), suggesting the interaction may be too tight to facilitate RNA tertiary folding and catalysis.

On a final note, while this work focused on amino acids, which occupy ~50% of the metabolites in E. coli cells and complex ~15 mM of the total 54 mM $Mg^{2+}$ in the E. coli cell, our findings have implications for the remaining ~40 mM bound $Mg^{2+}$ ions, some of which is almost certainly weakly chelated. Phosphate monoesters, such as sugar phosphate and nucleotide monophosphates also bind weakly with $Mg^{2+}$. These weakly bound $Mg^{2+}$ complexes may further promote RNA stability, folding, and catalysis.

## Methods

**Concentrations of cellular amino acid-magnesium complexes.** Interaction between glutamate, the most abundant amino acid in E. coli, and $Mg^{2+}$ in aqueous solution can be represented by the following equilibrium reaction,

$$Glu + Mg^{2+} \rightleftharpoons Glu \cdot Mg^{2+} \tag{1}$$

The binding affinity of glutamate for magnesium can be represented as

$$K_{D,Glu} = \frac{[Glu][Mg^{2+}]}{[Glu \cdot Mg^{2+}]} \tag{2}$$

From this, we can define the fraction of glutamate-chelated $Mg^{2+}$ as

$$f_{Glu \cdot Mg^{2+}} \equiv \frac{[Glu \cdot Mg^{2+}]}{[Glu]_0} \tag{3}$$

which can be shown to give

$$f_{Glu \cdot Mg^{2+}} = \frac{[Mg^{2+}]}{K_{D,Glu} + [Mg^{2+}]} \tag{4}$$

or

$$[Glu \cdot Mg^{2+}] = [Glu]_0 \frac{[Mg^{2+}]}{K_{D,Glu} + [Mg^{2+}]} \tag{5}$$

where $[Glu]_0$ is the total concentration of glutamate, which has been reported to be 96 mM in E. coli cells[12,13], $[Mg^{2+}]$ is the concentration of free magnesium in E. coli, and $[Glu \cdot Mg^{2+}]$ is the concentration of glutamate and magnesium complex. The $[Mg^{2+}]$ is ~2 mM in bacterial cells and ~0.5 mM in eukaryotic cells (Fig. 1)[6,9,10]. $[Mg^{2+}]$ in Eq. 4 can be substituted by the appropriate value between 0.5 and 2.0 mM.

Next, we consider a mixture of the five most abundant amino acids in E. coli (glutamate, aspartate, valine, glutamine, and alanine; Supplementary Fig. 1). Since the $K_D$ value of valine for magnesium is unknown, we removed it from later experiments and thus this calculation. Equilibria of the rest of three amino acids can be represented by following equilibria (Eqs. 6a, 6b, and 6c).

$$Asp + Mg^{2+} \rightleftharpoons Asp \cdot Mg^{2+} \tag{6a}$$

$$Gln + Mg^{2+} \rightleftharpoons Gln \cdot Mg^{2+} \tag{6b}$$

$$Ala + Mg^{2+} \rightleftharpoons Ala \cdot Mg^{2+} \tag{6c}$$

Thus, the binding affinity of each amino acid for $Mg^{2+}$ can be represented as

$$K_{D,Asp} = \frac{[Asp][Mg^{2+}]}{[Asp \cdot Mg^{2+}]} \tag{7a}$$

$$K_{D,Gln} = \frac{[Gln][Mg^{2+}]}{[Gln \cdot Mg^{2+}]} \tag{7b}$$

$$K_{D,Ala} = \frac{[Ala][Mg^{2+}]}{[Ala \cdot Mg^{2+}]} \tag{7c}$$

From this, we can define the fraction of each amino acid-chelated $Mg^{2+}$ as follows.

$$[Asp \cdot Mg^{2+}] = [Asp]_0 \frac{[Mg^{2+}]}{K_{D,Asp} + [Mg^{2+}]} \tag{8a}$$

$$[Gln \cdot Mg^{2+}] = [Gln]_0 \frac{[Mg^{2+}]}{K_{D,Gln} + [Mg^{2+}]} \tag{8b}$$

$$[Ala \cdot Mg^{2+}] = [Ala]_0 \frac{[Mg^{2+}]}{K_{D,Ala} + [Mg^{2+}]} \tag{8c}$$

where $[Asp]_0$, $[Gln]_0$, and $[Ala]_0$ are the total concentrations of aspartate, glutamine, and alanine, which have been reported to be 4.2, 3.8, and 2.6 mM in E. coli cells, respectively (Supplementary Fig. 1 and Supplementary Table 1)[12,13], and $[Asp \cdot Mg^{2+}]$, $[Gln \cdot Mg^{2+}]$, and $[Ala \cdot Mg^{2+}]$ are the concentrations of the appropriate amino acid-magnesium complex. The cellular abundance of each magnesium complex was calculated by using Eqs. (5), (8a), (8b), and (8c). The resulting abundances are summarized in Supplementary Table 1. Contribution of those four amino acid-chelated $Mg^{2+}$ given by (9).

$$[AA \cdot Mg^{2+}] = [Glu]_0 \frac{[Mg^{2+}]}{K_{D,Gln} + [Mg^{2+}]} + [Asp]_0 \frac{[Mg^{2+}]}{K_{D,Asp} + [Mg^{2+}]}$$
$$+ [Gln]_0 \frac{[Mg^{2+}]}{K_{D,Gln} + [Mg^{2+}]} + [Ala]_0 \frac{[Mg^{2+}]}{K_{D,Ala} + [Mg^{2+}]} \tag{9}$$

**Preparation of RNA transcripts.** Sequences of HDV-like ribozymes for drz-spur-3 and CPEB3 were obtained from previous reports[22,46]. DNA templates for these RNAs were prepared by the following oligonucleotide DNA sets, where "F" is for forward and "R" is for reverse, the underlined sequences correspond to the T7 promoter, and lower case shows start of transcription and 5′ leader sequences of the ribozymes:

drz-spur-3 F; GCG AAA TTT <u>AAT ACG ACT CAC TAT A</u>gc gcc aaa cat GAC ACT GAG TGA GAA ACG TCC CCG TCG TAG,

drz-spur-3 R; TAA TGT GAG AAT TGG CTA CGT TGA AAC AAC GCA TTA CCG ACA CTA CGA CGG GGA CGT TTC TCA C,

CPEB3 ribozyme F; GCG AAA TTT <u>AAT ACG ACT CAC TAT A</u>gg atc aag ggg ata aca GGG GGC CAC AGC AGA AGC GTT CAC GTC GCA GCC,

CPEB3 ribozyme R; CAG CAG AAT TCG CAG ATT CAC CAG AAT CTG ACA GGG GCT GCG ACG TGA ACG CTT CTG.

Transcription was performed in a reaction mixture of 40 mM Tris-HCl (pH 7.5), 2 mM DTT, 1 mM spermidine, 2.5 mM NTPs, 25 mM MgCl$_2$, and 20 ng/µL of template DNA at 37 °C for 4 h. To prepare cleaved drz-spur-3 for use in thermal denaturation experiments and ILP experiments, self-cleavage was promoted during transcription. The cleaved drz-spur-3 was purified by 10% denaturing PAGE (8.3 M urea) and recovered by ethanol precipitation. To prepare full-length internally labeled precursor CPEB3 ribozyme for self-cleavage assays, NTPs and MgCl$_2$ concentrations were reduced to 0.6 mM each and 4.4 mM, respectively, to increase internal labeling efficiency and decrease self-cleavage, and 10 µCi of fresh [α-$^{32}$P]-GTP (purchased from Perkin-Elmer) was added. Transcription was performed at 37 °C for just 2 h in the presence of 10 µM blocking oligonucleotide 5′-d(GTG GCC CCC TGT TAT C) to inhibit the self-cleaving reaction. Precursor RNA was then purified by 10% denaturing PAGE (8.3 M urea) and recovered by ethanol precipitation. The dry pellet was dissolved in water and stored at −20 °C. Specific activity was determined by liquid scintillation counting. Preparation of glmS ribozyme and hammerhead ribozyme 16 (HH16) were prepared as previously described[24,25].

**Monitoring of thermal denaturation by UV absorbance**. Before the experiment, buffer replacement was carried out. In brief, the RNA solution was applied to Amicon ultra centrifugal filter (MW 3000 cutoff) and centrifuged at 14,000× g at 4 °C for 10 min. Then, buffer for thermal denaturation was added into the filter and centrifuged in the same condition. This buffer replacement was repeated three times. Cleaved drz-spur-3 transcript (final concentrations of 0.5 OD/mL) was renatured at 95 °C for 3 min and cooled to room temperature for at least 10 min. Then, either 5 × glutamate-magnesium buffer, 5 × amino acids-magnesium buffer, 5 × EDTA-magnesium buffer, 5 × metabolite-magnesium buffer, or MgCl$_2$ solution was added and renatured at 55 °C for 3 min and cooled to room temperature for at least 10 min. The renatured samples were centrifuged by 14,000 rpm for 10 min to degas and remove debris. Thermal denaturation experiments were performed in the background of 10 mM sodium cacodylate (pH 7.0), 140 mM KCl, and various concentrations of Mg$^{2+}_{free}$ in the presence or absence of amino acid. In the case of 2 mM Mg$^{2+}_{free}$, 1 × glutamate-magnesium buffer contains 96 mM total glutamate and 13.3 mM total MgCl$_2$, where 2 mM is Mg$^{2+}_{free}$ and 11.3 mM is chelated Mg$^{2+}$ (Eq. 5) (Supplementary Table 1); 1 × amino acid-magnesium buffer contains 96 mM glutamate, 4.2 mM aspartate, 3.8 mM glutamine, 2.6 mM alanine, and 16.0 mM MgCl$_2$; and 1 × EDTA-magnesium buffer contains 11.3 mM EDTA and 13.3 mM MgCl$_2$. All buffer compositions are summarized in Supplementary Data 1. The resultant samples were analyzed by an OLIS spectrophotometer with a data point acquired every 0.5 °C and a heating rate of 0.5 °C/min at 230–300 nm. These analyses were conducted in duplicate. Melting points ($T_M$) were obtained from unfolding denaturation curves. Then, $T_M$ values were converted into the apparent relative changes in free energy using the following Eq (10).

$$\Delta\Delta G = \Delta H_0 \times (1 - T_{M0}/T_M) \quad (10)$$

where $\Delta H_0$ and $T_{M0}$ are the enthalpy and melting points obtained from the reference state. $\Delta H$ was assumed and confirmed to be relatively independent of condition.

**In-line probing analysis**. The cleaved drz-spur-3 was 5′-end labeled by T4 polynucleotide kinase (purchased from New England BioLabs) using [γ-$^{32}$P]-ATP (purchased from Perkin-Elmer). The $^{32}$P-5′ end labeled drz-spur-3 (230,000 cpm) was incubated with ILP reaction buffer containing 25 mM Tris-HCl (pH 8.3), 100 mM KCl, either 13.3 mM or 16.0 mM of MgCl$_2$, and either 96 mM glutamate, 106 mM amino acids or 11.3 mM EDTA in 10 µL of reaction mixture at 37 °C for 0 min, 24 h, and 47 h. The ILP reaction was stopped by addition of 2 × formamide buffer [95% formamide, 20 mM EDTA, 0.02% xylene cyanol, 0.02% bromophenol blue]. RNA alkaline ladder and RNase T1 ladder were prepared by following the Donis-Keller method with some minor changes[47]. A volume of 2 µL of the sample was loaded onto a 10% PAGE (8.3 M urea) sequencing gel and fractionated at 40 W for ~2 h. These experiments were performed in duplicate. The data at 0 and 24 h time points were analyzed by SAFA software[48].

**Self-cleaving assay**. Approximatley 20,000 cpm of $^{32}$P-internal labeled CPEB3 ribozyme and 5′-labeled substrates for glmS ribozyme and HH16 were used for self-cleaving assays. Before the reaction for the CPEB3 ribozyme and HH16, RNAs in reaction buffer containing 25 mM Tris-HCl (pH 7.5) and 140 mM KCl were renatured at 95 °C for 3 min and cooled to room temperature for 10 min. For self-cleaving assay in crowding condition, the CPEB3 ribozyme was renatured in reaction buffer containing 25 mM Tris-HCl (pH 7.5), 140 mM KCl and 20% PEG8000 at 55 °C for 3 min and cooled to room temperature for 10 min. The reaction was initiated by addition of MgCl$_2$ or amino acid-magnesium solutions that were pre-adjusted to pH 7.5. A 2 µL of aliquot was withdrawn at specific time intervals and quenched with an equal volume of 2 × formamide buffer. Samples were separated by 10% PAGE (8.3 M urea). Self-cleaving assays for the glmS ribozyme were performed with 0.25 nM 5′-labeled substrate and 100 nM enzyme strand in 25 mM Tris-HCl (pH 7.5), 140 mM KCl, and 10 mM GlcN6P. Before the

reaction, the enzyme and substrate were renatured at 95 °C for 3 min and snap cooled on ice for 10 min. The RNA was then added to the reaction mixture containing MgCl$_2$ or amino acid-magnesium solutions that were pre-adjusted to pH 7.5 and heated at 55 °C for 3 min and cooled at room temperature for 10 min. Following this, the reaction mixture was added to GlcN6P to initiate the reaction. A volume of 3 µL aliquots were withdrawn at specific time intervals and quenched with 2 × formamide buffer with 50 mM EDTA, and 1 mM heparin was present to prevent aggregation of RNA in gel wells. Samples were separated by 20% PAGE (8.3 M urea). The RNA fragments were visualized by a PhosphorImager (Typhoon 650; GE Healthcare) and then quantified by ImageQuant (GE Healthcare). Experiments were conducted four times (two biological repeats and two technical repeats) for CPEB3 ribozyme in the buffer condition of amino acid-chelated magnesium and in triplicate for CPEB3 ribozyme in the crowding condition, and the pH was checked with pH paper after completion of the reaction to ensure the additives did not affect the buffer capacity. The $R_P$, $S_P$ and dithio substrates were purified by HPLC according to our previous study[25]. Kinetics thio effect were performed by the same procedures as self-cleaving assay for glmS except that the sulfur-substituted substrates. Experiments on glmS and hammerhead were conducted at least three times.

Rates of self-cleavage were analyzed by nonlinear least-squares fitting. In the case of CPEB3 ribozyme, since internally labeled CPEB3 ribozyme was used for the reaction, three fragments were observed: precursor CPEB3 ribozyme, cleaved CPEB3 ribozyme, and 5′-leader RNA. Fraction cleaved was defined by the Eq (11).

$$f_{cleaved} = \frac{I_{cleaved} + I_{5'-leader}}{I_{precursor} + I_{cleaved} + I_{5'-leader}} \quad (11)$$

where $I_{cleaved}$, $I_{precursor}$, and $I_{5'-leader}$ are the band intensity of the cleaved CPEB3 ribozyme, the precursor CPEB3 ribozyme, and the 5′-leader RNA, respectively. The $f_{cleaved}$ vs time was plotted and fit to single exponential curve [Eq. (12)].

$$f_{cleaved} = A + B \cdot e^{-k_{obs}t} \quad (12)$$

where $A$ is the fraction of ribozyme cleaved at completion, $-B$ is the amplitude of the observable phase, $k_{obs}$ is the observed first-order rate constant for ribozymes self-cleaving for the non-burst phase, and $t$ is time. In the case of the two-pieces glmS and HH16, since 5′-labeled substrates were used, fraction cleaved was defined by Eq. (13).

$$f_{cleaved} = \frac{I_{cleaved}}{I_{intact} + I_{cleaved}} \quad (13)$$

where $I_{cleaved}$ and $I_{intact}$ are the band intensity of the cleaved 5′-fragment and the intact RNAs, respectively. The $f_{cleaved}$ vs time were plotted and fit to Eq. (12).

Effect of substitution of a sulfur atom for a non-bridging oxygen at a scissile phosphate group is referred to as a thio effect. The thio effect was determined as:

$$\text{Thio effect} = k_O/k_S \quad (14)$$

where $k_O$ and $k_S$ are the rate constants for the oxo and the sulfur-substituted substrates.

**Small-angle X-ray scattering (SAXS) data collection**. Data were collected at room temperature on beam line G1 at MacCHESS, the Macromolecular Diffraction Facility at the Cornell High Energy Synchrotron Source (CHESS). The detector used for data recording was a dual 100 K−S SAXS/WAXS detector (Pilatus). The sample capillary-to-detector distance setup allowed for simultaneous collection of small- and wide-angle scattering data, covering a broad momentum-transfer range ($q$ range) of 0.0075−0.8 Å$^{-1}$ ($q = 4\pi \sin(\theta)/\lambda$, where $2\theta$ is the scattering angle). The energy of the X-ray beam was 9.8528 keV (1.2584 Å), and the synchrotron X-ray beam diameter was 250 µm × 250 µm. Before all sample data collection, buffer replacement was carried out with the same procedures as thermal denaturation experiment, and no buffer mismatch was observed in the SAXS measurement. A volume of 40 µL of cleaved CPEB3 ribozyme (0.2 mg/mL) was renatured in 30 mM HEPES (pH 8.0) and 120 mM KCl (final concentration) at 95 °C for 3 min and cooled to room temperature for 10 min. A volume of 10 µL of the buffer containing MgCl$_2$ or chelated magnesium was added to the RNA solution and heated at 55 °C for 3 min and cooled at room temperature for 10 min. Final concentration of the measured sample is described in Supplementary Data 1. The RNA solution was centrifuged at 14,000 rpm for 10 min, and 30 µL of the RNA solution was used for the data collection. Thirty scattering images were collected per sample.

**SAXS data analysis**. The SAXS data analysis was performed according to our previous paper[25,36,40]. In brief, scattering curves were analyzed by BioXTAS RAW software[49]. The averaged scattering curves of the buffer were subtracted from the averaged scattering curves of the sample. The linear region of the ln ($I$) vs $q^2$ plot, wherein $q_{max}R_g$ < 1.3, was identified for each sample using the Guinier analysis, then the radius of gyration ($R_g$) and molecular weight of the sample were determined. Theoretical scattering curves were calculated from the crystal structure of HDV ribozyme (3NKB), which was adjusted to human CPEB3 ribozyme as shown

in our previous paper[36], using FoXS webserver[50,51] and then compared to the experimental scattering curves. The maximum particle dimension ($D_{max}$) was determined by GNOM, the ATSAS software package[52]. Ten individual DAMMIF bead models were created from GNOM output file and averaged by DAMA-VER[53,54]. The output bead model from DAMAVER was aligned with the crystal structure using SUPCOMB and compared in PyMOL[55].

**Data availability**. The data that support the findings of this study are available from the corresponding author upon reasonable request.

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

## Acknowledgements

We thank Dr. R. Gillilan and Dr. L. Pollack for help with the SAXS experiments and Dr. N.H. Yennawar and K. Leamy for assistance on the SAXS data analysis. This work is conducted at the Cornell High Energy Synchrotron Source (CHESS), which is supported by the National Science Foundation and the National Institutes of Health/National Institute of General Medical Sciences under NSF award DMR-0936384, using the Macromolecular Diffraction at CHESS (MacCHESS) facility, which is supported by GM-103485 from the National Institutes of Health, through its National Institute of General Medical Sciences. We thank K. Messina for assistance with fast hand-mixing reaction time points and members of Bevilacqua group for discussions and suggestions. This study was supported by NIH grant R01-GM110237. We are grateful to the Uehara Memorial Foundation for supporting R.Y.

## Author contributions

R.Y., J.L.B, E.A.F, and P.C.B. performed the experiments. R.Y. and P.C.B. designed the experiments, analyzed the data, and wrote the paper.

## Additional information

**Competing interests:** The authors declare no competing interests.

