## [Peer Review File · Nature Communications]

PEER REVIEW FILE

Reviewers' Comments:

Reviewer #1 (Remarks to the Author):

In this manuscript, Bevilacqua and coworkers investigated that effects of weakly chelate Mg^{2+} inducing amino acids (aaCM) on the ribozyme activity and stability. The aaCM enhance the stability, cleavage activity of several ribozymes and decreased K_d values of Mg^{2+} for the ribozyme active folding. This manuscript has been improved and they have taken the referee's comments. The discussion for effects of molecular crowding and cytoplasm on ribozymes is well done and clearly supported their conclusion.

However, this referee feels the additional experiments should support the author's conclusion. For example, aaCM effects on maximum yield for the cleaved RNAs for the ribozymes are still unclear. As author wrote that the maximum yields for CPEB-3 and glmS ribozyme in 0.1 mM Mg^{2+} was increased by GluMg, however, that of hammerhead ribozyme was clearly decreased. Generally, the maximum yield for the cleaved RNAs is depended on the correct folding of the active structure for the ribozyme. The author should carry out additional experiments to reveal the difference of the maximum yield among the ribozymes in the presence of aaCM such as RNA foot printing and assay for ribozyme activity depending on the reaction temperatures and discuss in more detail.

Thus, this referee feel that this manuscript will be acceptable for publication after the revision.

Reviewer #2 (Remarks to the Author):

Yamagami et al describe the effects of co-solutes such as glutamate on the folding equilibria of RNA and, by extension, ribozyme self-cleavage activity.

The revised manuscript addresses many of the specific referee comments, and the technical presentation of the results has improved. For example, they now clarify that anions such as glutamate mainly act by weakly binding Mg^{2+} ions in competition with the RNA. Under such conditions, the pool of hydrated “free” Mg^{2+} ions is low (1-2 mM), but the total ion concentration available to the RNA is in fact much higher (10-15 mM), thereby allowing the RNA to fold. I draw the authors’ attention to descriptions of metal ion buffers, which have been neglected in recent times but may prove even more useful for RNA folding studies (eg, D. Perrin, Buffers for pH and Metal Ion Control, Chapman & Hall, 1974). The authors also clarify that crowding agents increase ribozyme activity by stabilizing the folded RNA, as previously reported by a number of authors. The solutions and sample preparation are now described more thoroughly.

The potential impact of this study on the RNA folding field is mixed. On the one hand, the simple effect of competing equilibria on metal ion-RNA interactions has been overlooked. By calling attention to these effects, this study is likely to spur others to consider such competing equilibrium when evaluating how RNAs fold inside the cell. On the other hand, the effects of ion equilibria have been described, as have the effects of crowding on RNA folding. RNA biochemists routinely account for the effects of metal ion chelators such as EDTA and nucleotide triphosphates in their experiments, even if they don’t always think about how this plays out in the cell – maybe everyone is used to seeing the emperor naked? The experiments of Yamagami et al are technically solid, but do not go beyond the very thorough and well controlled studies on metal ion-nucleic interactions performed in the 1970’s.

Minor points:

In response to a referee comment, the authors revised the text on pg. 3 to explain that “free” Mg^{2+} means Mg^{2+} ions that are not bound to anything. This is worse, in my view. The use of “free” to describe ions that are not chelated by an organic solute is sloppy; the statement that such ions are not bound to anything is just plain wrong. At a minimum, the “free” ions are hydrated by waters!

Please make sure that every figure legend states which RNA was used for the experiment reported in the figure.

The outmoded and incorrect term “prokaryotes” should be replaced with “bacteria” and “archaea” as appropriate.

The authors should consider using the term “in cell” rather than “in vivo”, because the latter is often taken to mean in the whole animal.

Reviewer #3 (Remarks to the Author):

The manuscript clearly demonstrates that many ligands (like amino acids) bind Mg^{2+} ions less tightly than RNA does. Whether this deserves a full experimental exploration in order to demonstrate is unclear to me. However, the experiments are well executed and the interpretations well supported by the data, and by first principles with respect to known or estimated binding affinities.

We thank the referees for careful reviewing and their helpful comments, which enable us to further improve our manuscript. Here are our point by point responses.

Referee #1 (Remarks to the Author):

In this manuscript, Bevilacqua and coworkers investigated that effects of weakly chelate Mg^{2+} inducing amino acids (aaCM) on the ribozyme activity and stability. The aaCM enhance the stability, cleavage activity of several ribozymes and decreased K_d values of Mg^{2+} for the ribozyme active folding. This manuscript has been improved and they have taken the referee's comments. The discussion for effects of molecular crowding and cytoplasm on ribozymes is well done and clearly supported their conclusion.

However, this referee feels the additional experiments should support the author's conclusion. For example, aaCM effects on maximum yield for the cleaved RNAs for the ribozymes are still unclear. As author wrote that the maximum yields for CPEB-3 and glmS ribozyme in 0.1 mM Mg^{2+} was increased by GluMg, however, that of hammerhead ribozyme was clearly decreased. Generally, the maximum yield for the cleaved RNAs is depended on the correct folding of the active structure for the ribozyme. The author should carry out additional experiments to reveal the difference of the maximum yield among the ribozymes in the presence of aaCM such as RNA foot printing and assay for ribozyme activity depending on the reaction temperatures and discuss in more detail.

RESPONSE: The reviewer seems to have misread our figure. Our comparison involves constant free Mg^{2+} of 0.5 mM. The maximum yield for hammerhead ribozyme increases ~4-fold in 0.5 mM Mg^{2+} (there are no data for 0.1 mM Mg^{2+} with hammerhead ribozyme). In Supplementary Figure 5 panel b, compare closed red boxes to open black boxes. In summary, the maximum yield for all ribozymes in the presence of gluCM consistently increases in low Mg^{2+} conditions.

Thus, this referee feel that this manuscript will be acceptable for publication after the revision.

Referee #2 (Remarks to the Author):

Yamagami et al describe the effects of co-solutes such as glutamate on the folding equilibria of RNA and, by extension, ribozyme self-cleavage activity.

The revised manuscript addresses many of the specific referee comments, and the technical presentation of the results has improved. For example, they now clarify that anions such as glutamate mainly act by weakly binding Mg^{2+} ions in competition with the RNA. Under such conditions, the pool of hydrated "free" Mg^{2+} ions is low (1-2 mM), but the total ion concentration available to the RNA is in fact much higher (10-15 mM), thereby allowing the RNA to fold. I draw the authors' attention to descriptions of metal ion buffers, which have been neglected in recent times but may prove even more useful for RNA folding studies (eg, D. Perrin, Buffers for pH and Metal Ion Control, Chapman & Hall, 1974). The authors also clarify that crowding agents increase ribozyme activity by stabilizing the folded RNA, as previously reported by a number of authors. The solutions and sample preparation are now described more thoroughly.

The potential impact of this study on the RNA folding field is mixed. On the one hand, the simple effect of competing equilibria on metal ion-RNA interactions has been overlooked. By calling attention to these effects, this study is likely to spur others to consider such competing equilibrium when evaluating how RNAs fold inside the cell. On the other hand, the effects of ion equilibria have been described, as have the effects of crowding on

RNA folding. RNA biochemists routinely account for the effects of metal ion chelators such as EDTA and nucleotide triphosphates in their experiments, even if they don't always think about how this plays out in the cell – maybe everyone is used to seeing the emperor naked? The experiments of Yamagami et al are technically solid, but do not go beyond the very thorough and well controlled studies on metal ion-nucleic interactions performed in the 1970's.

RESPONSE: We thank the referee for their opinion on the impact of our work. We do indeed discuss competition of Mg^{2+} between RNA and amino acids (see p15). Also, we measure EDTA- Mg^{2+} effects and discuss Mg^{2+} interaction with sugar phosphate and nucleotide monophosphates. What is surprising about our findings is that the relatively weak binding of Mg^{2+} to glutamate promotes several key RNA functions: it improves RNA folding, it limits RNA degradation, and it enhances ribozyme activity, up to 100,000-fold. The vast majority of RNA folding, binding, and catalysis studies in the literature consider either high Mg^{2+} concentrations (of ≥ 10 mM) or low Mg^{2+} concentrations (of ≤ 2 mM). The former strongly promote RNA folding, while the latter have been reported to be more reflective of *in vivo* folding. However, our study reports a third scenario: low free Mg^{2+} plus high amino acid-chelated Mg^{2+} . This scenario, which is perhaps most reflective of *in vivo* conditions, displays properties more similar to the RNA function-promoting conditions of high free Mg^{2+} .

Minor points:

In response to a referee comment, the authors revised the text on pg. 3 to explain that “free” Mg^{2+} means Mg^{2+} ions that are not bound to anything. This is worse, in my view. The use of “free” to describe ions that are not chelated by an organic solute is sloppy; the statement that such ions are not bound to anything is just plain wrong. At a minimum, the “free” ions are hydrated by waters!

RESPONSE: Yes, free ions are coordinated to water and we have changed this text to “fully hydrated by water at the inner-coordination sites on Mg^{2+} ”.

Please make sure that every figure legend states which RNA was used for the experiment reported in the figure.

RESPONSE: We have added RNA species on Figure 1, 5, and Supplementary Figure 6.

The outmoded and incorrect term “prokaryotes” should be replaced with “bacteria” and “archaea” as appropriate.

RESPONSE: We have corrected throughout the manuscript.

The authors should consider using the term “in cell” rather than “in vivo”, because the latter is often taken to mean in the whole animal.

RESPONSE: We agree and have changed “*in vivo*” to either “in cell” or “cellular”.

Referee #3 (Remarks to the Author):

The manuscript clearly demonstrates that many ligands (like amino acids) bind Mg^{2+} ions less tightly than RNA does. Whether this deserves a full experimental exploration in order to demonstrate is unclear to me. However, the experiments are well executed and the interpretations well supported by the data, and by first principles with respect to known or estimated binding affinities.

RESPONSE: We thank the reviewer for encouraging comments. Our experimental studies are justified by the RNA function-specific results obtained in the presence of amino acid chelated Mg^{2+} , which could not have been easily predicted in advance, including enhancement of folding, stability, and catalysis.